# Unraveling the Keratin Expression in Oral Leukoplakia: A Scoping Review

**DOI:** 10.3390/ijms25115597

**Published:** 2024-05-21

**Authors:** Guru Murthy O, Jeremy Lau, Ramesh Balasubramaniam, Agnieszka M. Frydrych, Omar Kujan

**Affiliations:** UWA Dental School, The University of Western Australia, Nedlands, WA 6009, Australia; 23121846@student.uwa.edu.au (G.M.O.); 21584322@student.uwa.edu.au (J.L.); ramesh.balasubramaniam@uwa.edu.au (R.B.); agnieszka.frydrych@uwa.edu.au (A.M.F.)

**Keywords:** oral, leukoplakia, cytokeratin, prognosis, biomarkers

## Abstract

Intermediate filaments are one of three polymeric structures that form the cytoskeleton of epithelial cells. In the epithelium, these filaments are made up of a variety of keratin proteins. Intermediate filaments complete a wide range of functions in keratinocytes, including maintaining cell structure, cell growth, cell proliferation, cell migration, and more. Given that these functions are intimately associated with the carcinogenic process, and that hyperkeratinization is a quintessential feature of oral leukoplakias, the utility of keratins in oral leukoplakia is yet to be fully explored. This scoping review aims to outline the current knowledge founded on original studies on human tissues regarding the expression and utility of keratins as diagnostic, prognostic, and predictive biomarkers in oral leukoplakias. After using a search strategy developed for several scientific databases, namely, PubMed, Scopus, Web of Science, and OVID, 42 papers met the inclusion and exclusion criteria. One more article was added when it was identified through manually searching the list of references. The included papers were published between 1989 and 2024. Keratins 1–20 were investigated in the 43 included studies, and their expression was assessed in oral leukoplakia and dysplasia cases. Only five studies investigated the prognostic role of keratins in relation to malignant transformation. No studies evaluated keratins as a diagnostic adjunct or predictive tool. Evidence supports the idea that dysplasia disrupts the terminal differentiation pathway of primary keratins. Gain of keratin 17 expression and loss of keratin 13 were significantly observed in differentiated epithelial dysplasia. Also, the keratin 19 extension into suprabasal cells has been associated with the evolving features of dysplasia. The loss of keratin1/keratin 10 has been significantly associated with high-grade dysplasia. The prognostic value of cytokeratins has shown conflicting results, and further studies are required to ascertain their role in predicting the malignant transformation of oral leukoplakia.

## 1. Introduction

Intermediate filaments (IF) are one of three polymeric networks found in eukaryotic cells. They are 10 nm in diameter and composed of numerous proteins encoded by various genes [1]. Intermediate filaments can be categorized into several subtypes based on the proteins that constitute them [2], namely, Class I (keratins a), Class II (keratins b), Class III (vimentin, desmin, and more), Class IV (nestin and more), Class V (lamins), and Class VI (filensin and more) [3]. Each protein is expressed in various cell types. For example, vimentin is found in mesenchymal cells, and nestin is found in glial cells [3]. Class I and Class II IF proteins are found in epithelial cells and were originally catalogued and termed cytokeratins by Moll in 1982 [4]. Schweizer further refined this in 2006 [5]; whereby he largely appreciated Moll’s system [4], but reclassified cytokeratins into the now recommended nomenclature of keratin groups. For human epithelium, these were categorized as human type I epithelial keratins, which include K9–K28, and human type II epithelial keratins, which include K1–K8 and K71–K80 [5].

At least one member of type I or type II keratin must be present to form the heteropolymeric structure that makes up the IF. This pairing leads to a highly stabilized polymer compared to its monomer form, which is easily broken down. This keratin pairing has been found to be varied within the same cell family type, between different cell types, tissues, and more. Consequently, they fulfil a wide range of functions, including maintaining cell structure, cell integrity, cell adhesion, cell growth, cell proliferation, and cell migration [3]

Primary keratins that are produced consistently in normal oral epithelium include K4/K13, which are found suprabasally in the non-keratinizing oral epithelium and K1/K10, which are found suprabasally in the keratinizing oral epithelium [6,7,8,9,10,11]. K5/K14 are found in the basal layer of both th keratinizing and non-keratinizing oral epithelium [4,6,8]. Another notable keratin co-expressed suprabasally is K76 (previously K2p), which has been found in the gingiva and palate, both keratinized oral epithelium [9,12]. K8, K18, and K19 have been found localized to Merkel cells in the oral epithelium [4]. K19 is also found in the basal cells in non-keratinizing mucosa [6,13]. K20 has been found to be a marker of taste buds and Merkel cells as well [14].

Of particular interest in the oral context is oral leukoplakia (OLK). The World Health Organization (WHO) defines OLK as a predominantly white plaque of questionable risk, having excluded (other) known diseases or disorders that carry no increased risk of cancer [15]. It is classified as an potentially malignant oral disorder [16]. Prominent histological features of OLK include hyperplasia with hyperkeratosis (ortho or para), with or without dysplasia [16]. Notably, there have been multiple studies that have reported altered keratin expression in OLKs [17,18,19].

Given the above, in combination with keratins role in the carcinogenic process and its antigenic stability [20], keratins remain an untapped source of potential in the diagnosis and management of OLKs. This is notably seen in a variety of other cancers, such as renal cell carcinomas, breast adenocarcinomas [21], salivary gland malignancies [22], and metastatic cancers [23]. The utility of biomarkers in oncology can be diagnostic, prognostic, or predictive. A diagnostic biomarker may be used to identify the presence of disease and also subtyping (e.g., cancer type). A prognostic biomarker identifies patients’ disease progression or recurrence (e.g., cancer prognosis) with or without treatment. A predictive biomarker identifies patients who are more likely to benefit from a particular treatment [24].

In the context of keratins, the utility of keratins in OLKs may be diagnostic; for example, they may be used as an adjunct for the detection grading of dysplasia. Keratins may be used as a prognostic tool, for example, in predicting malignant transformation (MT). They may also be utilized as a predictive tool with which to identify the ideal treatment [25]. Hence, this manuscript aims to review the current knowledge of keratins expressed in OLK and their potential roles (e.g., diagnostic, prognostic, or predictive). This is the first scoping review to unravel the expression of keratins in oral leukoplakia and evaluate their diagnostic, prognostic, and predictive values.

## 2. Methods

The protocol was registered at the Open Science Framework, (registration https://doi.org/10.17605/OSF.IO/P2VXS) (accessed on 5 April 2024).

### 2.1. Eligibility Criteria

Eligibility criteria are outlined in Table 1 as guided by PECOS (participants, exposure, comparators, and study designs) based on the aims of this scoping review. As such, all relevant studies were filtered based on these predefined criteria, resulting in select studies being included in this scoping review.

### 2.2. Data Sources and Search Strategy I

Two authors (GO and JL) independently searched the following electronic databases: PubMed, OVID, SCOPUS, and Web of Science. No lower date limits were set, and the upper date limit was set to February 2024. The latest search was conducted on the 7th of February 2024. To increase the sensitivity of search results, searches in MeSH and free terms were utilized. The core search strategy included ‘Cytokeratin’ + ‘Location’ + ‘Pathology’, whereby location included the various sites in the oral cavity (buccal mucosa, tongue, palate, gingiva, etc.) that was searchable in the respective database, and pathology referred to leukoplakia or dysplasia or carcinoma in situ. Fields of search included ‘Title’, ‘Abstract’, ‘Keywords’, and ‘Author-specified keywords’, which were modified for each search database accordingly. A complete script of the search strategy in various databases is included in Appendix A. Endnote X9.3.3 was utilized to manage references and duplicates. The retrieved records were also manually scanned by the authors after endnote removed duplicates to ensure all duplicates were removed.

### 2.3. Study Selection and Screening I

Two investigators (GO and JL) independently evaluated the articles retrieved from the databases. The first round of evaluation was performed by reviewing the title and abstract of the studies. The remaining studies were then considered suitable for the final round of eligibility assessment, which involved reading the full text, after which the final list of eligible studies remained (Figure 1). The reference list of eligible studies was also scanned for eligible studies. The included studies were cross-referenced between the authors at every stage, and any disagreements were resolved by further review and discussion among three reviewers (GO, JL, and OK).

### 2.4. Data Extraction and Synthesis

Two reviewers (GO and JL) independently retrieved data from the included studies. A senior reviewer (OK) supervized the process and verified the extracted information. Data extracted were compiled using a standardized method using Microsoft Excel v. 365. Domains collected included authors, year, title, DOI, main keratin investigated, study’s aim, type of study, tissue site, storage of the sample, methodology for assessment of keratin, statistical analysis employed, dysplasia grading system, and study findings for 20 studies individually as a pilot round. After discussing with all 3 reviewers, data items collected included author, year, tissue of interest and site, controls, dysplasia grading system, if statistical analysis was completed, methodology of analysis of keratin expression, and study type (which included expression, diagnostic, prognostic, or predictive). Expression refers to the study being aimed to investigate the expression patterns and or changes in expression of keratin for that particular condition. This may have included increase/decrease in protein staining intensity, increase/decrease in number of cells stained. and increase/decrease in samples stained for that particular keratin. These were independently retrieved by 2 reviewers (GO and JL), and a senior reviewer (OK) verified the extracted information. Some studies included additional ‘non-keratin’ biomarkers, cell line studies, and non-oral sites. Only the relevant data consistent with the inclusion and exclusion criteria were extracted.

Results were synthesized and presented on a keratin pair (if applicable) or on a per-keratin basis. Overall, these were divided into keratin protein studies and mRNA studies. Within each keratin/keratin pair group, results were divided into the type of study (e.g., expression, prognosis). The general trend within each group was based on the highest number of studies supporting a particular pattern, for which studies against this trend were also discussed. Additional pertinent data for each trend were also discussed when required.

## 3. Results

A total of 689 studies were found, of which 121 duplicates were removed. Records were screened based on titles and abstracts, whereby 488 were removed. Eighty studies were assessed for eligibility (Figure 1). Of these, 42 studies fulfilled the inclusion and exclusion criteria, and 1 additional article was included, which was found by searching the reference list of the articles [26,27,28,29,30,31,32,33,34,35,36,37,38,39,40,41,42,43,44,45,46,47,48,49,50,51,52,53,54,55,56,57,58,59,60,61,62,63,64,65,66,67,68] (Table 2). The year of publication ranged from 1989 to 2024. A total of 2461 tissue samples from leukoplakia with or without dysplasia were compared to approximately 352 normal tissues/controls (some studies did not specify the number of normal specimens used as controls). Sample sizes varied from as low as 6 [31] to 200 [40] in the OLK/dysplasia group, with an average of 57 samples per study (median = 40). As many as 17 out of the 43 studies did not perform statistical testing. Forty-two studies utilized hematoxylin and eosin (H&E) and immunohistochemistry (IHC), and one study only used mRNA analysis without H&E [58]. In addition to H&E and IHC, seven studies [29,30,39,58,60,61,68] used in situ hybridization (ISH) to detect mRNA expression in their samples, and one study [68] used utilized reverse transcriptase polymerase chain reaction (RT-PCR) and electrophoresis to analyze mRNA expression levels between their groups. Additionally, three studies used gel electrophoresis and immunoblotting [33,57,63]. Seven different dysplasia grading systems were represented. Nine studies [31,32,33,38,41,42,43,52,56] used WHO 1978 [69], two studies [55,68] used WHO 1997 [70], seven studies [27,36,45,48,50,53,62] used WHO 2005 [71], five studies [28,40,54,65,66] used WHO 2017 [72], two studies [36,67] used SIN system [71], two studies [46,61] used Kramer 1980 [73], and one study [64] used Grassel-Pietrusky and Hornstein 1982 [74] (Table 2). Six studies [26,29,30,34,39,58] graded dysplasia without providing details of the criteria used, and seven studies [35,37,44,49,51,59,60] identified the presence of dysplasia without providing the details of the criteria used. Three studies did not assess for dysplasia at all [47,57,63]. Most studies utilized semi-quantitative assessment of staining intensity (Table 2). Two studies utilized automated systems to quantify IHC staining [37,52], and nine studies analyzed IHC concerning the various layers of the epithelium [27,31,34,42,43,46,51,62,64].

All 43 studies explored various keratins and their expression in OLK and dysplasia, and 5 investigated their prognostic potential with regard to MT [26,28,65,66,67]. No studies evaluated keratins as a diagnostic adjunct or predictive tool. Keratins 1–20 are represented in the included studies (Table 2).

Regarding the prognostic studies, the first of which [51] investigated K4/K13 and had a follow up time ranging from 1 to 173 months (mean of 69 months and median of 61 months). The second prognostic study [67] investigated K13 and K17 and had a follow-up time ranging from 1 to 155 months (with a mean of 50.4 months and no median reported). The third study [65] investigated K13 and K17, with a follow up time of 11–183 months for patients who had MT of their OLK (median 51 months) and 109–258 months (median 148 months) for their non-progressors. The fourth study [66] investigated K13 and K17 as well, with a follow up time of 12–300 months (median 63 months). The fifth study [28] investigated K13 and K17, with a follow up time of 4–290 months (median 52 months).

## 4. Discussion

This scoping review explored the keratin expression in OLKs and their potential utility. Despite the heterogeneity of the studies, general trends of keratin expression can be observed. It must be emphasized that these are trends, and conflicting results of various studies are presented in Table 3, with direct comparisons being quite difficult. Nonetheless, these trends can be broadly summarized as follows (Table 3):Loss of expression of primary keratins normally found in oral epithelium. Specifically, K1/K10 and K4/K13 suprabasally in keratinized and non-keratinized epithelium. This is also true for K2p for keratinized mucosa. This suggests that dysplasia disrupts the terminal differentiation pathway of primary keratins.Suprabasal extension of primary keratins of the basal layer, which include K5/K14 and K19. However, K19 may also express atypically (suprabasal extension) in inflammatory lesions.The gain of K17 expression in OLK samples, which is the most studied keratin with respect to the increase in protein expression. Aside from the above, K8 and K18 have also shown increased expression, albeit in a smaller proportion of samples compared to K17.

None of the studies have managed to investigate the effectiveness of keratin as a diagnostic utility; that is, the improvement of inter- and intra-observer reliability with the utility of keratin is needed. This has been recently achieved in sites involving the upper aerodigestive tract, whereby the authors utilized the suprabasal extension of K19 to improve inter- and-intra observer reliability for the diagnosis of dysplasia [75]. The closest study in this review to achieving this was Becker [28], who reportedly used K13 and K17 to improve diagnosis of dysplasia. However, data on intra- or inter-observer variability improvements were not provided.

Nonetheless, from this review, further studies on the following keratins may be helpful in the following scenarios (high-grade vs. low-grade dysplasia):LOE of K1/K10.
There is a marked loss of K1/K10 expression in severe dysplasia (regardless of original site of OLK samples, e.g., keratinized or non-keratinized).
Suprabasal extension of K19 to the most superficial layers of epithelium indicates severe or high-grade dysplasia.Prognostic utility: GOE of K17.Retention of K13 expression may indicate a lower risk of MT.Further studies investigating the significance of GOE for K6, K7, K8, K16, and K18 in OLK. Current evidence shows that not many studies stain for these keratins in their samples, and as such, data are not hampered by the quality but rather by quantity of studies.

Additionally, there were some notable exclusions from our review as they did not fit the inclusion and exclusion criteria. Five of these had samples which were taken adjacent to OSCC samples, which are at risk of exhibiting a different genetic signature and possibly phenotype to traditional OLKs [17,18,19,76,77]. One was non-specific for keratin antibodies [78], and for another two, the methodology was not consistent with the aims of the paper (e.g., aiming to prove a new entity or validate a new technique [79,80]). Most of these studies found results similar to the trends found. However, of particular interest was Khanom’s study [19], which found a K15 decreasing expression in basal cells of dysplastic samples as the grade of dysplasia worsened, similar to Sakamoto’s study [53]. However, they also had hyperplasia samples, for which K19 showed altered expression (suprabasal extension). In contrast, K15 largely retained its staining, prompting the authors to conclude that it was a more stable keratin to utilize as a diagnostic adjunct compared to K19.

Furthermore, it must be noted that most studies focused on the protein expression of keratins, likely due to their ease of staining. However, it may be beneficial to analyze mRNA expressions (Table 4) as the protein expression may be suppressed by post-transcriptional mechanisms. As such, the detection of mRNA may yield more accurate results.

It is interesting to note that with regards to gain of expression, the K17 has the most evidence in the context of OLK which is also reflected in the wider literature regarding OSCC. K17 has been extensively researched in a variety of cancers and has established oncogenic roles [81]. Squamous cell carcinomas of the oral cavity have been shown to have increased expression of K17, with some clinical studies finding prognostic implications for OSCC [82]. Several authors have attempted to investigate the mechanistic role of K17 in oncogenesis via in vitro OSCC cell line studies. Mikami [83] found that in ZK-1 cell lines with K17 knockout, the loss of keratin did not affect cell migration or invasion but decreased in cell size as compared to normal controls with retained K17. In Khanom’s in vitro study, they found that K17 stimulated Akt/mTOR pathway and glucose uptake, which supports its role in tumor growth. In Mikami’s 2017 study [84], knockout of K17 increased the number of cleaved caspase-3-positive HSC-2 cells, which are involved in apoptosis. As such, their conclusion advised that K17’s oncogenic role in OSCC tumor growth may not only be through the Akt/mTOR pathway but also the suppression of apoptosis.

The main limitation of this study is the lack of quantitative synthesis attributed to the significant heterogeneity between studies reviewed, even when assessing the state of knowledge for one keratin. Every study had a slightly varying methodology, comparison groups (or lack thereof), staining techniques, source of monoclonal antibodies, assessment protocols, populations, and more (Table 2). As such, the level of evidence was difficult to ascertain, hence the publication of a scoping review on this emerging topic. Ultimately, prospective cohort studies assessing keratin as a prognostic or predictive biomarker would yield the highest evidence. Inter- and intra-observer comparison studies would help with its diagnostic utility.

However, many questions still remain with respect to the utility of keratins in OLKs. In which context should the biomarker be used? Should it be used to distinguish between dysplasia and hyperkeratosis? Should it be used to aid in the differentiation of various grades of dysplasia? What instrument/tool/methodology will be used to assess keratins? Is keratin best assessed through the percentage of positive cells or through staining intensity and assessed by layers? The keratins and their suggested utility, as outlined in this review, require further research for validation.

## 5. Conclusions

The use of keratin as a biomarker in OLKs has untapped potential. Further research is necessary to fully elucidate the roles keratins may play as, to date, most studies have only focused on the expression of keratins in OLKs, with limited studies investigating the diagnostic, prognostic, and predictive utility of keratins. The current state of knowledge suggests that the loss of expression of primary keratins K4/K13, K1/K10, and K2p/K76 suprabasally can be investigated for their use as a diagnostic adjunct. The suprabasal spread of K19 and LOE of K15 basally may also have potential use as a diagnostic adjunct. The superficial spread of K19 may also be used to distinguish between high-grade and low-grade dysplasia. Furthermore, retention of K13 in OLK samples without dysplasia may have prognostic potential. There is evidence that the gain of expression of K17 may have some prognostic utility. The significance of protein expression of K8, K18, K6, and K16 in a small number of OLK samples should also be explored as there is a lack of studies investigating these keratins.

## Figures and Tables

**Figure 1 ijms-25-05597-f001:**
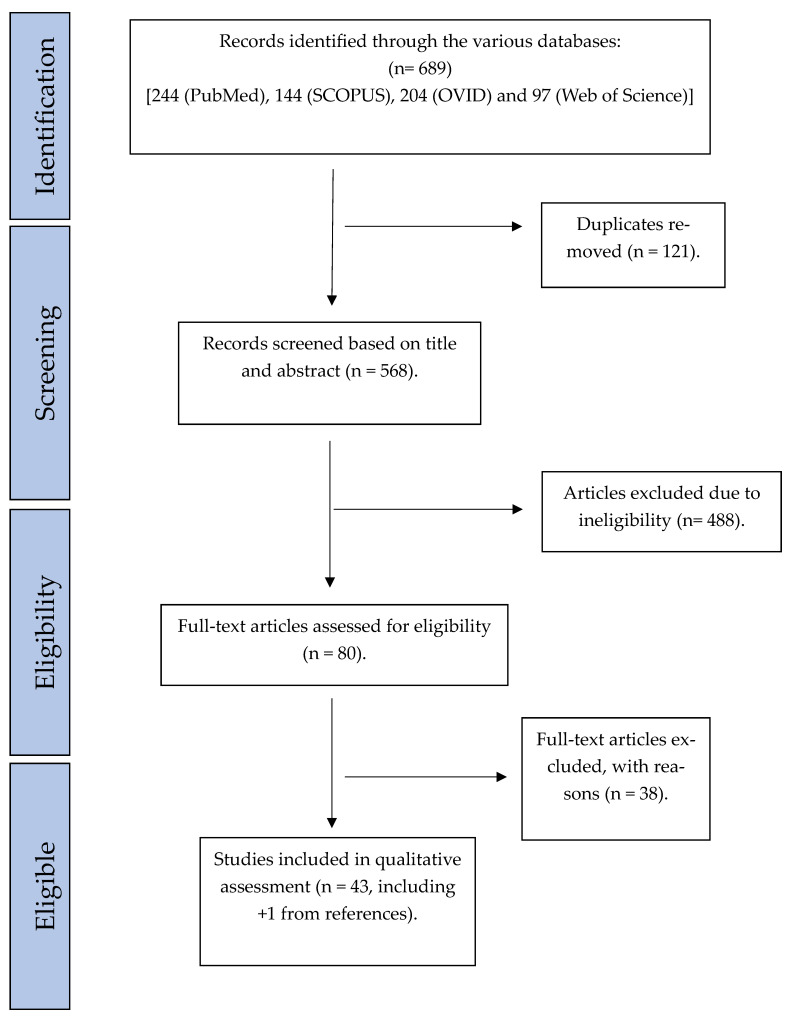
Flowchart representing systematic literature search.

**Table 1 ijms-25-05597-t001:** PECOS criteria for inclusion and exclusion of studies.

Criteria	Inclusion Criteria	Exclusion Criteria
**Participants**	Incisional or excisional biopsies of oral leukoplakia with or without oral epithelial dysplasia. Human tissues only (including all genders, races, and ages) Lesions from the following sites: tongue, alveolus, gingiva, buccal mucosa, retromolar areas, floor of mouth, hard palate, and soft palate.	Leukoplakia and/or dysplasia cannot be excised from the margins of an oral squamous cell carcinoma. Liquid or brush biopsies.
**Exposure**	Assessment of keratins via any methodology (i.e., histochemistry, immunohistochemistry, polymerase chain reaction, etc.). Methodology needs to have utilized techniques that enable identification of a single keratin protein or gene.	
**Comparison**	Oral mucosal tissues with no clinical signs and symptoms of dysplasia or inflammatory diseases. Oral mucosal tissues with clinical signs and symptoms of inflammatory disease. Non dysplastic OLK. Dysplastic OLK. OLK with various dysplastic grades.	-
**Outcome**	Loss of expression of keratin. Gain of expression of keratin. Expression patterns of keratin. Malignant transformation. Treatment outcomes.	
**Studies**	Original studies only. Studies only in English.	

**Table 2 ijms-25-05597-t002:** Overview of studies on keratin expression in oral leukoplakia.

Author/Year	Keratin of Interest	Tissue of Interest and Site	Normal Controls (If Any)	Dysplasia Grading System and Statistical Analysis (SA).	Analysis of CK Expression	Study Type
Ambatipudi et al., 2013 [26]	K76 (K2p)	61 total: hyperplasic lesions with focal mild–moderate dysplasia Site not specified	35 normal tissues from the gingiva-buccal tissues and 7 inflamed tissues not associated with oral malignancy or premalignant lesions	Grading system not specified No grading or detection of the absence/presence of dysplasia in their leukoplakia (OLK) samples SA completed, however not specifically between normal controls and OLK	Hematoxylin and eosin (H&E) with immunohistochemistry (IHC) immunostaining and qRT-PCR K76 protein expression—semi quantitative (0 = no staining; +1 = weak staining in <10% of cells; +2 = moderate staining and/or 10–50% of positive cells; +3 = strong staining in more than 50% of cells by pathologist For statistical analysis, stained tissues were categorized into two groups (0 and +1) and (+2 and +3).	Expression
Barakat et al., 2015 [27]	K15	30 total: 10 mild, 10 moderate, and 10 severe, and 10 with oral hyperkeratosis Site not specified	5 normal controls Site not specified No ST completed	WHO (2005) [72] Dysplasia graded No SA completed	H&E and IHC Semiquantitative 3 layers identified (stratum basale, spinosum, and corneum) Intensity was graded as mild (<25% cells stained), moderate (25–50% cells stained), and strong (>50% of cells stained) in the areas showing most intense staining at 200× magnification.	Expression
Becker et al., 2024 [28]	K13 and K17	131 total: 52 differentiated dysplasia, 42 keratosis without dysplasia, 14 mild dysplasia, 12 moderate dysplasia, and 11 severe dysplasia Sites: vestibular, buccal, tongue, and floor of mouth (FOM)	No controls	WHO (2017) [73] Dysplasia graded SA completed	H&E and IHC. Keratin expression labelled as ‘normal’(positive) or ‘aberrant’ (negative); K13 loss and K17 expression (even partially) was considered aberrant	Expression and prognosis
Bloor et al., 2001 [29]	K4, K13, K1, and K10	23 total: 9 mild, 7 moderate, and 7 severe from non-keratinized sites	6 normal controls Site from the BM	Grading system not specified Dysplasia graded No SA completed	H&E, IHC, and ISH	Expression
Bloor et al., 2003 [30]	K2e, K1, and K10	13 total: 6 Keratosis with no dysplasia, 2 mild dysplasia, 3 moderate dysplasia, and 2 severe dysplasia Sites from buccal mucosa (BM), lingual mucosa dorsal tongue, lateral tongue, ventral tongue, floor of mouth (FOM), labial mucosa, and lateral tongue soft palate	6 normal controls Sites from BM ventral tongue and gingiva	Grading system not specified Dysplasia graded No SA completed	H&E, IHC, and PCR (−) no immunostaining for protein (+/−) <5% cells positive (+) 10–30% cells positive (++) 40–70% cells positive (+++) 80–100% cells positive	Expression
Cema et al., 998 [31]	K10, K10/13,K5/6, K17, K18, and K19	6 total: 1 mild dysplasia, 2 moderate dysplasia, 2 severe, and 1 carcinoma in situ Sites from: FOM, BM, ventral and lateral surface of tongue	Normal oral mucosa, epithelial hyperplasia, and atrophy Numbers not discussed Site from FOM, BM, ventral and lateral surface of tongue	WHO (1978) [65] Dysplasia graded No SA completed	H&E and IHC Epithelium was divided into stratum basal, stratum suprabasal (spinosum and granulosum) and stratum superficial, semi-quantitatively measured using as follows: (−) no expression (1+) <5% positive cells (2+) 5–20% cells (3+) 21–50% (4+) 51–80% cells (5+) >80% cells	Expression
Cintorino et al., 990 [32]	K1 and K19	20 total: 14 cases of OLK with no dysplasia. Sites from BM vestibular mucosa, FOM, soft palate, gingiva, and dorsal tongue 6 cases of OLK with mild to severe dysplasia Sites from vestibular mucosa, soft palate, hard palate, pilastrum, and dorsal tongue.	Normal controls presented in results, but not discussed in materials and methods Site and number of controls not specified	WHO (1978) [65] Dysplasia graded No SA completed	H&E and IHC Semiquantitative: (−) negative (−/+) weak and/or irregular (+) strong and regular	Expression
Ermich et al., 989 [33]	K1, K2, K3, K4, K5,K6/K11, K7/K13, K8, K9, K10, K12, K14/15, K16, K17, K18, and K19	20 total: 4 degree I dysplasia, 2 degree II dysplasia, and the remining 14 samples had no dysplasia 14 from non-keratinizing and 6 from keratinizing sites	31 normal controls in total. 10 from non-keratinized mucosa, 12 from keratinized tissues (8 gingiva and 4 hard palate) and 9 from specialized mucosa (5 from lip and 4 dorsal tongue)	WHO (1978) [65] Dysplasia graded No SA completed	H&E and SDS PAGE and MABs to keratins	Expression
Farrar et al., 2004 [34]	AE1, AE3, and K14	40 total: 10 samples from 4 tissue types: oral tissues showing benign oral lesions (squamous papillomas) or inflammatory changes; oral tissue displaying evidence of mild, moderate, and severe dysplasia Site not specified	10 normal controls Site not specified	Grading system not specified. Dysplasia graded No SA completed specifically for expression of keratins	H&E and IHC Qualitative analysis of intensity and location of staining (basal layer, lower 1/3, middle 1/3, and upper 1/3) Intensity of staining graded at 0, + (1–50 +ve cells), ++ (51–150 +ve cells), and +++ (>150 +ve cells)	Expression
Farrukh et al., 2015 [35]	K13	37 total: 21 hyperplastic, 16 severely dysplastic tissue Sites from BM, tongue, FOM, and lip	19 normal controls Site from BM	Grading system not specified However, presence/absence of dysplasia was identified; dysplasia was not graded SA completed	H&E and IHC. Qualitative analysis using different categories, including the following: strong diffuse +ve, extensive or near complete loss, complete loss with focal reactivity in keratin pearls, complete loss with focal strong reactivity in keratin pearls, complete loss with focal weak reactivity in keratin pearls, loss except in superficial granular layer, complete loss	Expression
Fillies et al., 2007 [36]	K5/K6, K8/18, K1, K10, K14, and K19	140 total: 117 OLK with no dysplasia, 23 leukoplakia (OLK) with dysplasia Site not specified	No controls	Squamous intraepithelial neoplasia (SIN) system [72] Dysplasia graded SA completed	HE and IHC Positive cells in each core (0 = no expression; 1 = >1% positive expression)	Expression
Gires et al., 2006 [37]	K8	8 OLK total Site not specified	9 normal controls Site not specified	Grading system not specified However, presence/absence of dysplasia was identified; dysplasia was not graded No SA completed	H&E, IHC, and AMIDA system (autoantibody-mediated identification of antigens)	Expression
Heyden et al., 992 [38]	K10, K13, and K14	30 OLK total Site from FOM and ventral surface of tongue	10 autopsy specimens of non-keratinized normal mucosa, from forensic cases 8–24 h after death caused by myocardial infarction or accident, and 5 biopsies from buccal mucosa containing metaplastic keratinized epithelium with no dysplastic changes were examined	WHO (1978) [65] Dysplasia graded SA completed	H&E and IHC Differences in staining intensity between cell layers of normal mucosa, dysplastic lesions, and carcinomas; keratin staining was recorded twice by the same investigator (AH), applying a semi-quantitative scoring system (negative (−), heterogeneous (−/+), homogenously intense (+++)) for each cell layer	Expression
Ida-Yonemochi et al., 2012 [39]	K13 and K16	23 total CIS samples from the tongue	10 foci of normal controls Site not specified	Grading system not specified Dysplasia graded No SA completed	H&E, IHC, RT-PCR, and ISH	Expression
Ikeda, 2020 [40]	K13 and K17	200 total: Dysplasia samples include 9 from the palate, 60 from the gingiva, 99 from the tongue, and 32 from the buccal mucosa	Normal controls used Site and sample size not discussed	WHO (2017) [73] Dysplasia graded SA completed	H&E and IHC IHC marker patterns for K13 (opposite true for K17): 1 = Normal expression pattern 2 = Heterogenous positivity with high intensity 3 = Heterogenous positivity with low intensity 4 = No positivity as a homogenous decrease with a border separating the normal area	Expression
Kannan et al., 994 [41]	K 10/11, K19, K18, and K14	60 total: 30 non-dysplastic OLK, 15 mild, 8 moderate, 7 severe All from non-keratinizing sites	10 normal controls 5 from keratinizing and another 5 from non-keratinizing sites	WHO (1978) [65] Dysplasia graded SA completed	H&E and IHC	Expression
Kannan et al., 994 [42]	K10 and K11	60 total: 30 non-dysplastic OLK, 15 mild, 8 moderate, 7 severe All from non-keratinizing sites	10 normal controls 5 from keratinizing and another 5 from non-keratinizing sites	WHO (1978) [65] Dysplasia graded SA completed	H&E and IHC Qualitative analysis—intensity of staining in basal, lower, and upper third; intensity rated from 0–6	Expression
Kannan et al., 996 [43]	K10/11, K13, K14, K16, K19, and K18	60 total: 30 non-dysplastic OLK, 15 mild, 8 moderate, 7 severe All from non-keratinizing sites	5 normal controls Site from non-keratinizing epithelium	WHO (1978) [65] Dysplasia graded SA completed	H&E and IHC Grade of staining (negative: 0, mild: 2, moderate: 4 and intense: 6) and layers (basal, lower spinal, upper spinal) were also scored.	Expression
Kiani et al., 2020 [44]	K13 and K17	85 dysplasia total Site not specified	No normal controls	Grading system not specified However, presence/absence of dysplasia was identified; dysplasia was not graded SA completed	H&E and IHC. Case was considered negative or positive depending on the intensity of staining of the cells; 10 high power fields in each slide were evaluated for K13 and K17 positivity	Expression
Kitamura et al., 2012 [45]	K17 and K13	108 total: 74 OLK without dysplasia and 34 OLK with dysplasia Site not specified	10 normal controls Site not specified	WHO (2005) [72] Dysplasia graded SA completed	H&E and IHC	Expression
Lindberg and Rheinwald, 1989 [46]	K19	19 OLK total Site from 9 FOM, 4 ventral tongue, 3 retromolar pad, 3 gingivae	Normal controls from keratinizing and non-keratinizing mucosa; did not specify number	Kramer (1980) [74] Dysplasia graded SA completed	H&E and IHC K19 expression was recorded for its presence or absence, in basal and/or suprabasal layers; no quantification of staining	Expression
Nanda et al., 2012 [47]	K8 and K18	10 OLK total Site from BM	10 normal controls Site from the BM	Grading system not specified Presence/absence of dysplasia was not identified; dysplasia was not graded SA completed	H&E and IHC The intensity of staining of epithelium (basal and suprabasal) was assessed as on a 4-point scale from -ve to +++ intense	Expression
Nobusawa et al., 2014 [48]	K13, K14, and K17	146 total: 43 mild, 63 moderate, and 40 CIS Site from tongue, gingiva, BM, FOM, hard palate, and soft palate	21 normal epithelial foci from surgical excisions Sites from tongue, gingiva, BM, and FOM	WHO (2005) [72] Dysplasia graded SA completed	H&E and IHC Staining for each keratin was assessed as positive when more than 10% of cells showed intracytoplasmic staining	Expression
Okada et al., 2010 [49]	K13, K14, and K17	30 total: 10 hyperkeratosis with no dysplasia, 10 hyperkeratosis with dysplasia, and 10 CIS Site from tongue	No controls	Grading system not specified However, presence/absence of dysplasia was identified; dysplasia was not graded No SA completed	H&E and IHC Qualitative analysis of intensity of staining—strong, intermediate, weak, negative	Expression
Rajeswari et al., 2021 [50]	K19	40 total: 10 cases of hyperplasia 10 mild, 10 moderate, and 10 severe Site not specified	Normal controls. Number and sites not specified	WHO (2005) [72] Dysplasia graded SA completed	H&E and IHC Sections initially scanned at lower power, if positive, ×40 zoom used for three microscopic fields; Allred score used for scoring; system as follows: Proportion score: 0 = no cells + 1 = ≤1% cells + 2 = 1–10% of cells + 3 = 11–33% of cells + 4 = 34–66% of cells + 5 = >66% of cells + Intensity score: 0 = none 1 = weak 2 = intermediate 3 = strong	Expression
Ram Prassad, 2005 [51]	K19	18 total: 6 mild, 6 moderate, and 6 severe dysplasia Site not specified	6 normal controls 4 non-keratinized and 2 keratinized sites	Grading system not specified However, presence/absence of dysplasia was identified; dysplasia was not graded SA completed	H&E and IHC 500 cells counted at 40× magnification in 3 layers—basal, suprabasal and spinous layer	Expression
Safadi et al., 2010 [52]	K19	43 total: 23 mild, 8 moderate, and 12 severe Site not specified	Polypoid fibrous hyperplasia to represent normal oral epithelium Site and number of controls not specified	WHO (1978) [65] Dysplasia graded SA completed	H&E and IHC Staining for K19 measured via ImageJ computer program—measured using color deconvolution plug in.	Expression
Sakamoto et al., 2011 [53]	K1, K2e, K4, K5, K6, K7, K8, K9, K10, K13, K14, K15, K16, K17, K18, K19, K20, and hair keratins	100 total Site not specified	Normal cells adjacent to OSCC specimens; for IHC, normal epithelium adjacent to OSCC and OED (100 sample) Sites not specified	WHO (2005) [72] Dysplasia graded No SA completed	H&E and IHC Assessment by comparing immunoreactivity in the lesion with that in normal epithelium of the same specimen	Expression
Sanguansin et al., 2021 [54]	K17	91 total: 33 OLK without dysplasia and 58 OLK with dysplasia Site not specified	12 normal controls Site not specified	WHO (2017) [73] Dysplasia graded SA completed	H&E and IHC Number of positive cases in a group calculated and also staining intensity graded on 0–3 score depending on number of cells that stained positive (semiquantitative—0 = ≤5% of immunoreactive cells; 1 = 6–25%, 2 = 26–50% and 3 = ≥51%).	Expression
Sawant et al., 2014 [55]	K1, K5, K8, and K18	52 OLK total Site from BM	10 normal controls Site from the BM	WHO (1997) [66] Dysplasia graded SA completed	H&E and IHC. IHC was quantified by visual assessment under x200 magnification; three fields (1 field = 100 cells) were counted by two pathologists Immunoreactivity was graded as follows: −/no = <10%; +/low = 11–30%; ++/moderate = 31–50%; +++/intense = >51%	Expression
Schaaij-Visser et al., 2010 [56]	K4 and K13	48 OLKs total Site not specified	No controls	WHO (1978) [65] Dysplasia graded SA completed	H&E and IHC semi quantitative—estimated % of stained cells x staining intensity (0 = absent; 1 = weak; and 2 = strong).	Expression and prognostic
Schulz et al., 992 [57]	K1, K2, K3, K4, K5, K6/11, K7/13, K8, K9, K10, K12, K14–15, K16, K17, K18, and K19	20 OLK total Site not specified	22 normal controls 10 from non-keratinized and 12 from keratinized gingiva	Grading system not specified Presence/absence of dysplasia was not identified; dysplasia was not graded No SA completed	H&E and IHC, SDS page, and Western blotting	Expression
Shahabinejad et al., 2021 [58]	K7 and K20	38 dysplasia total Site not specified	No controls	Grading system not specified Dysplasia graded SA completed	H&E and qRT-PCR	mRNA expression
Sihmar et al., 2022 [59]	K8 and K18	30 OLK total Site not specified	10 normal controls Site not specified	Grading system not specified However, presence/absence of dysplasia was identified; dysplasia was not graded SA completed	H&E and IHC Semi-quantitative assessment: the number of K8- and K18-positive cells was counted in 50 cells of each field by two observers; the scoring was as follows: (−) no color; (+) yellow; (++) light brown; and (+++) dark brown Cases were assigned to one of the following categories: 0% positive cells (−); 10% positive cells (+); 10–25% positive cells (++); 26–50% positive cells (+++); or more than 50% positive cells (++++)	Expression
Su et al., 994 [60]	K7, K8, and C18	9 severe dysplasia total Sites not specified	10 normal controls Site from BM, labial FOM hard palate dorsum of tongue and gingival sites	Grading system not specified Dysplasia was graded; however, all dysplasia samples was combined into 1 group for analysis No SA completed	H&E, IHC, and ISH	Expression
Su et al., 996 [61]	K14 and K19	9 moderate-to-severe dysplasias Site from non-keratinized sites	10 normal controls Site from BM, labial, FOM, hard palate, dorsum tongue, and gingival sites	Kramer (1980) [74] Dysplasia was graded However, all dysplasia samples was combined into 1 group for analysis No SA completed	H&E, IHC, and ISH	Expression
Takeda et al., 2006 [62]	K19	62 total: 10 hyperplasia, 10 mild dysplasia, 10 moderate, 13 severe, and 10 CIS, 9 two phase dysplasia classified as moderate dysplasia Sites from tongue, gingiva, BM, hard palate, FOM, and lip.	10 normal controls Sites from tongue, gingiva, BM, hard palate, FOM and lip	WHO (2005) [72] Dysplasia graded No ST completed for keratins	H&E and IHC Qualitative—assessed via layers—BM, parabasal layer (two layers above BM and next to basal layer, supra basal layer—above the parabasal layer) using microscope at ×400 zoom	Expression
Vaidya et al., 998 [63]	K1/K2, K4, K5, K6, K7, K8, K10, K11, K13, K14, K16, K17, and K18	20 OLK total Site from buccal mucosa	Normal controls taken from adjacent to three of the OLK samples	Grading system not specified Presence/absence of dysplasia was not identified; dysplasia was not graded. No SA completed	Gel electrophoresis and immunoblotting for K1–19	Expression
Vigneswaran et al., 989 [64]	K1, K2, K5, K7, K8, K10, K11, K18, and K19	150 OLK total Site not specified	15 normal controls Sites from labial, BM, lingual, gingival, and palatal tissue	Grassel-Pietrusky and Hornsein (1982) [75] Dysplasia graded No SA completed	H&E and IHC Semiquantitative 0 to +4 with regard to extent (1/3 s) of epithelium involved	Expression
Wils et al., 2020 [65]	K13 and K17	84 OLK total Site not specified	8 normal controls Site not specified	WHO (2017) [73] with the addition of differentiated dysplasia. Presence/absence of dysplasia was identified. Dysplasia was not graded. SA completed	H&E and IHC	Expression and prognostic
Wils et al., 2023 [66]	K13 and K17	176 OLK total Site not specified	No controls	WHO (2017) [73] with the addition of differentiated dysplasia. Dysplasia graded SA completed	H&E and IHC K13 was negative when staining was even partly absent K17 was scored positive when epithelium showed detectable levels	Expression and prognostic
Yagyuu et al., 2015 [67]	K13 and K17	94 total High-grade dysplasia (HGD) and low-grade dysplasia (LGD) Site not specified	No controls	WHO (2005) [72] and their expanded SIN system Dysplasia graded SA completed	H&E and IHC K13: 0 = strong diffuse expression; 1 = weak or patchy expression; 2 = no expression. The reverse of this scale was used for K17 Discrepant slides were re-evaluated using dual vision microscope to achieve a consensus	Expression, and prognostic
Yoshida et al., 2015 [68]	K14 and K19	17 total: 9 LGD and 8 HGD Site not specified	15 normal controls from Tissue from surgical margins	WHO (1997) [66] Dysplasia graded SA completed	H&E and IHC and RT PCR. >500 epithelial cells in each slide—percentage of +ve cells calculated in addition with the mean for all 44 samples; values were used as labelling indices	Expression

Legend: CIS—carcinoma in situ; FOM—floor of mouth; GOE—gain of expression; H&E—hematoxylin and eosin; HGD—high-grade dysplasia; IHC—immunohistochemistry; ISH—in situ hybridization; LGD—low-grade dysplasia; LOE—loss of expression; OLK—oral leukoplakia; OSCC—oral squamous cell carcinoma; qRT-PCR—quantitative reverse transcription polymerase chain reaction.

**Table 3 ijms-25-05597-t003:** General trends of keratins protein in oral leukoplakia.

Keratin	Studies That Investigated the Keratins of Interest	Expression in Oral Leukoplakia (OLK)	Overall Impressions for Utility
K1 and/or K10	12 studies in total: Bloor [29] Bloor [30] Cema [31] Cintorino [32] Ermich [33] Fillies [36] Heyden [38] Kannan [41] Kannan [42] Sakamoto [53] Sawant [55] Vaidya [63]	General trend: There appears to be a slight gain of expression of K1/K10 in OLK with no dysplasia and mild dysplasia before they are progressively lost in severe dysplasia [29,38,41,42]. This initial gain of expression (GOE) in cases with no dysplasia and mild dysplasia is interesting and may reflect a reactionary process to mild frictional or genetic insult at the early stages, producing primary keratins before they are progressively lost and replaced by other keratins as the lesion increases in genetic aberrations. This occurred irrespective of non-keratinized or keratinized sites. All studies were statically analyzed except for Bloor’s papers. Studies that partially support the general trend: Sawant [55] had 82% of mild, 71% of moderate, and 100% of severe positivity for K1. The initial decrease in expression from mild and moderate is consistent with the general trend mentioned above. However, 100% expression in the severely dysplastic group is inconsistent with the trend of K1 loss of expression (LOE) in severe dysplasia. However, in this study, there were only three samples of severe dysplasia. Cintorino [32] found a decrease in K1 expression in OLKs with dysplasia derived from keratinized tissue vs. OLKs without dysplasia. The opposite was true if the OLK was present in non-keratinized tissue. However, although the presence of dysplasia was assessed, it was not graded and as such, the loss of K1 in severe dysplasia was not investigated in this study. Studies that do not support the general trend: Sakamoto [53] found increased expression of K1/K10 in dysplastic samples compared to normal controls. However, there was no indication of the site of their dysplasia samples and normal controls, nor did they differentiate the various stages of dysplasia. No statistical analysis was completed either. Fillies [36] found no relationship with regards to K1/K10 expression between OLK without dysplasia and OLK with dysplasia. However, their samples were not graded for dysplasia, the sites of their samples were not mentioned, and there were no control groups. Bloor [30] concluded there was an upregulation of K1/K10 in dysplastic samples compared to controls. However, they only had two samples for mild, moderate, and severe dysplasia and six normal controls. The result table was varied with regards to expression, and as the sample size was so low, no statistical analysis was completed. There was minimal (<5% of cells) K10 expression in both normal and dysplastic samples in Cema’s [31] paper. As such, no differences were noted. There was minimal-to-no expression of K10 in normal and OLK samples in Ermich’s [33] study; hence, no differences were found. K1 expression was maintained when comparing keratinized and specialized mucosa to OLK samples. Non-keratinized normal controls had no expression of K1. Vaidya [63] did not find K10 expression in their normal controls (non-keratinized) or OLK samples. K1 was not considered as immunoblotting when combined with K2.	Loss of K1/K10 is most significant in severe dysplasia as compared to mild or moderate cases, whereby in some cases (especially in non-keratinized mucosa), there may be an upregulation early on. As such, its use as a diagnostic adjunct is possibly to aid in distinguishing high-grade dysplasia (HGD) from low-grade dysplasia (LGD), whereby the latter would retain the staining. However, further studies are required to validate this.
K2p (K76) and K2e	4 studies in total: Ambatipudi [26] Bloor [30] Ermich [33] Sakamoto [53] Schulz [57]	K2p: General trend of LOE of K2p (K76) in dysplastic samples. Number of samples positive for K2p expression decreased between normal controls (100%) to OLK with dysplasia (44%) in Ambatipudi’s study [26]. No significant pattern could be detected between the controls and OLK samples in Ermich’s study [33]. K2e: General trend of GOE of K2e in dysplastic samples. K2e is not normally thought to be found in the oral epithelium. However, Bloor [25,30] found extremely weak protein expression in the suprabasal layers of the gingiva and lateral tongue. In reactive and dysplastic, hyperkeratotic samples, K2e protein was expressed intensely under orthokeratinized layers. There was minimal to no expression in parakeratinized epithelium. Sakamoto [53] found no K2e in their normal controls but weak expression suprabasally in 2/10 of their dysplastic samples. Normal control sites are not mentioned. Uncertain: Schulz [57] had no K2 (did not clarify if K2p/76 or K2e) expression in normal non-keratinized mucosa and 1/12 normal controls positive in keratinized mucosa. In dysplastic samples, there were 4/14 samples positive for K2 in non-keratinized sites and 3/6 samples positive in samples from keratinized sites, indicating a GOE in dysplasia. No statistical analysis was completed.	LOE of primary keratin K2p/K76 as a diagnostic adjunct for the detection of OLK with dysplasia. Further studies would be required to validate this. Furthermore, it would be worth investigating if there are changes to K2p expression in reactionary or inflammatory lesions to assess its usefulness in differentiating keratosis as a reaction (i.e., frictional keratosis) or an OLK sample. Additionally, the GOE of K2e in some dysplastic samples is an unexpected finding. Its significance as a diagnostic and prognostic biomarker is worth exploring.
K3	1 study in total: Ermich [33]	Ermich [33] found that between 25 and 50% of their OLK samples were positive for K3. In their normal controls, K3 was not found in non-keratinized mucosa but was found in all samples of keratinized mucosa and 3–9 samples of specialized mucosa.	K3 is a keratin normally found in corneal epithelium of the eye. It is not considered to be a keratin found normally in the oral epithelium. Its detection in normal and OLK samples in Ermich’s study is interesting, especially given that in the same study, its keratin pair K12 was not detected in either normal or controls. Furthermore, Emrich’s study has a low sample size, and no SA was completed. The significance of this GOE is likely limited.
K4 and/or K13	17 studies in total: Becker [28] Bloor [29] Ermich [33] Farrukh [35] Heyden [38] Ida Yonemochi [39] Ikeda [40] Kiani [44] Kitamura [45] Nobusawa [48] Okada [49] Sakamoto [53] Schaaij Visser [56] Vaidya [63] Wils [65] Wils [66] Yagyuu [67]	**Expression studies:** General trend: Even with the large heterogeneity between all 17 studies investigating K4 and/or K13, 13 of these studies unanimously show decreased K4 and K13 expression with advancing stages of dysplasia. Studies that do not support the general trend: Only Ermich’s [33] and Vaidya [63] study suggested otherwise, whereby there was no difference between protein expression of K4 [33,63] and K13 [63] between normal controls and OLK samples. However, the major flaw in these studies is the method of assessment, namely, gel electrophoresis (SDS-PAGE) with immunoblotting, which provided a categorical outcome (e.g., absolute yes or no for the presence of keratin). There was no quantification process, and as such, the experiment itself had no capability to report on any GOE or LOE. **Prognostic studies:** General trend: Loss of K4 [56] and K13 [28,56,65,66,67] did not predict MT. Additional details: However, in Wils’ first paper [65], retention of K13 was significantly associated with the absence of MT in cases with no morphological dysplasia; that is, OLK without dysplasia with retained K13 staining was significantly associated with the absence of MT. Becker [24] found that cases with LOE of K13 had a shorter MT time in all their entire cohort and subgroups (OLK with dysplasia, OLK without dysplasia, differentiated dysplasia) compared to those who retained it.	LOE of primary keratin K4 and K13 as a diagnostic adjunct for the detection of OLK with dysplasia. Further studies would be required to validate this. Furthermore, it would be worth investigating if there are changes to K4/K13 expression in reactionary or inflammatory lesions to assess its usefulness in differentiating keratosis as a reaction (i.e., frictional keratosis) or an OLK sample. The prognostic studies have provided varied results, but most suggest that loss of K4 and particularly K13 did not predict MT. However, K13 loss may indicate a shorter MT, and K13 retention in OLK may be protective of MT, and as such, it may be utilized in other ways, such as less-intensive surveillance [66]. Larger, multicentre cohort studies would be required to unravel the behavior of lesions that lose expression of K4 and K13.
K5 and/or K14	12 studies in total: Cema [31] Ermich [33] Farrar [34] Fillies [36] Heyden [38] Kannan [41] Okada [49] Sakamoto [53] Sawant [55] Su [61] Vaidya [63] Yoshida [68]	General trend: Suprabasal spread of K5 [31,53] and K14 [38,41,49,53,61,68] in dysplastic samples. With regards to protein expression, the results are mixed. Five studies [41,49,53,61,68] studies support increased K14 protein expression with worsening grades of dysplasia. One study [38] supports decreased K14 protein expression with worsening grades of dysplasia. Studies that do not support the general trend: Farrar [34] did not find any statistically significant results for K14 (LOE or GOE), even when comparing the various dysplastic grades. However, their normal samples contained a high intensity of K14 in all layers (K14 should only be expressed in the basal layer in normal epithelium), which questions the internal validity of their study. Fillies [38] found no protein expression of K14 in their OLK samples and OLK samples with dysplasia. As such, there were no significant results. K5 was not considered as the MAB stained for both K5 and K6. Sawant [55] found increased numbers of samples with LOE of K5 with worsening dysplasia (i.e., mild to severe). Vaidya [63] found a LOE of K5 and K14 of their OLK samples compared to controls. No significant differences could be detected between the controls and OLK samples in Ermich’s study [33], as both controls and OLK samples retained K5 expression.	Heterogenous results; however, suprabasal extension of K14 past the basal and parabasal layers can be used as a diagnostic adjunct to identify the presence of dysplasia. Extent of spread cannot be used to diagnose level of dysplasia. K5/K14 expressions in reactionary and inflammatory processes need to be further evaluated.
K6	3 studies in total: Ermich [33] Sakamoto [53] Vaidya [63]	K6 mildly expressed in suprabasal sites in Sakamoto’s normal controls [53]. Strong expression suprabasally in 6/10 of their dysplastic controls. K6 is not expressed in normal or OLK samples in Vaidya’s study [63]. No significant differences could be detected between the controls and OLK samples in Ermich’s study [33], as both controls and OLK samples retained K6 expression.	Limited studies conducted on K6. Nonetheless, it appears that a small number of samples can gain the expression of K6. Further studies will be required to elucidate the significance of this GOE.
K7	3 studies in total: Sakamoto [53] Su [60] Vaidya [63]	General trend: A small amount of OLK +/− dysplasia samples gain expression of K7. Sakamoto [53] did not find K7 protein expression in their normal controls or dysplasia samples. Su’s study [60] found no K7 protein was found in their normal samples. In their dysplasia samples, K7 protein was found in 5/9 samples. Vaidya [63] found no expression of K7 in in normal non-keratinized epithelium and only 2/20 OLK lesions expressed K7.	Based on the limited data available, there seems to be a GOE of K7 protein in some samples. The significance of these findings are unclear, and as such, they would require further studies to investigate and perhaps validate its use as a biomarker.
K8 and or K18	12 studies in total: Cema [31] Ermich [33] Fillies [36] Gires [37] Nanda [47] Sakamoto [53] Sawant [55] Schulz [57] Sihmar [59] Su [60] Vaidya [63] Vigneswaran [64]	General trend: A small amount of OLK (with or without dysplasia) expressed a GOE of K8 and/or K18 when compared to their normal controls [36,37,47,55,57,60,63]. Most studies found no expression of K8 and/or K18 in their OLK samples and normal controls or no significant/remarkable differences between the two groups [33,57,64].	Heterogenous results. Very poor quality of studies, with only for studies completing statistical analysis [37,48,56,60]. GOE of K8 and/or K18 is not found in all OLK samples, and as such, it would be a poor diagnostic adjunct. However, its utility as a prognostic or predictive biomarker for the samples that do express these keratins is worth exploring further.
K9	3 studies in total: Ermich [33] Sakamoto [53] Schulz [57]	No expression of K9 in normal or OLK samples.	K9 is typically only thought to be found in the epidermis of palmar and plantar surfaces. The results of these studies reflect this. It is unlikely that K9 has much significance in the context of OLK and dysplasia.
K12	2 studies in total: Ermich [33] Schulz [57]	K12 is not found in normal controls or OLK samples.	K12 is typically only thought to be found in the corneal epithelium of the eye. The results of these studies reflect this. It is unlikely K12 has much significance in the context of OLK and dysplasia.
K15	3 studies in total: Barakat [27] Sakamoto [53] Vaidya [63]	K15 was found in the basal cells of normal controls in Barakat’s [27] study. They found this expansion into suprabasal layers in OLK samples with dysplasia; however, their results table does not support this. Additionally, no statistical testing was completed. K15 was expressed strongly in the basal cells of normal controls in Sakamoto’s study [53] and weakly in the basal cells of 8/10 of their dysplasia samples. K15 was not found in normal controls or OLK samples in Vaidya’s study [63].	Limited and poor quality of data available for K15. All studies had small sample sizes, and no statistical analysis was completed. Further studies are required to confirm its presence in the basal layer and its alterations in OLK and dysplastic tissues.
K16	4 studies in total: Ermich [33] Ida-Yonemochi [39] Sakamoto [53] Schulz [57] Vaidya [63]	General trend: GOE of K16 in a small proportion of OLK or dysplasia samples. No expression found in normal controls in Ida-Yonemochi’s [39] study. K16 protein was detected in 7/23 OLK samples suprabasally. K16 was mildly expressed in the suprabasal layers of normal controls in Sakamoto’s study [53]. As many as 6/10 dysplastic samples had strong expression of K16 suprabasally. No expression of K16 in normal controls in Vaidya’s [63] study. As many as 2/20 OLK samples expressed K16. Studies that do not support the general trend: All normal controls from keratinized mucosa (8/8) and 50% of samples from non-keratinized mucosa (5/10) had K16 protein expression in Schulz study [57]. All OLK samples from keratinized mucosa (6/6) and just over 50% of OLK samples in non-keratinized mucosa (8/14) had K16 expression. No pattern could be identified. No significant differences could be detected between the controls and OLK samples in Ermich’s study [33] as both controls and OLK samples retained K16 expression.	Limited and poor quality of data available for K16. All studies had small sample sizes, and no statistical analysis was completed. The studies which showed no or minimal expression of K16 in their normal tissues showed a GOE in some samples. The studies in which K16 was expressed in normal tissues showed no change. The utility and interpretation of K16 is currently impeded by the lack of quality and quantity of studies. Further studies are required to explore the role of K16.
K17	15 studies in total: Becker [28] Cema [31] Ermich [33] Ikeda [40] Kiani [44] Kitamura [45] Nobusawa [48] Okada [49] Sakamoto [53] Sanguansin [54] Schulz [57] Vaidya [63] Wils [65] Wils [66] Yagyuu [67]	**Expression studies:** General trend: Generally, all studies show a GOE of K17 OLK samples (+/−dysplasia) compared to normal controls [28,44,45,48,49,53,54,57,65,66,67]. Some studies found an overall increase in expression in severe grades [40,45,48,67]. Some showed an initial increase in mild, followed by decreasing expression in moderate and severe grades [49,54]. Studies that do not support the general trend: Cema’s study [31] and Vaidya’s study [63] did not detect K17 in their normal or OLK samples. However, their samples were too small for statistical analysis. Ermich [33] found K17 in some samples of normal controls and OLK. No pattern of change was noted. Moreover, sample sizes were too small for statistical analysis. **Prognostic studies:** Three studies found that GOE of K17 did not predict MT of OLKs [65,66,67]. Becker’s paper [28] found those with GOE of K17 and a higher rate of MT with a shorter transformation time.	There appears to be a GOE of K17 in significant samples of both OLK with dysplasia and without dysplasia. As such, it may be useful as a diagnostic adjunct when trying to distinguish a reactive lesion form OLK. However, none of the authors included samples in which a reactive process (frictional keratosis) was used as controls. However, the significance of this GOE is still unclear as two prognostic studies have found that it does not influence the risk of MT, while another found an increased risk with the GOE of K17. As such, larger, multicentre cohort studies would be required to truly elucidate its prognostic potential.
K19	15 studies in total: Cema [26] Cintorino [27] Ermich [28] Fillies [31] Kannan [36] Lindberg [41] Rajeswari [45] Ram prassad [46] Safadi [47] Sakamoto [48] Schulz [52] Su [56] Takeda [57] Vaidya [58] Yoshida [63]	General trend: Most papers support the trend for a GOE of K19 as dysplasia progressed [32,33,36,41,46,50,51,52,53,57,62,63,68], irrespective of whether K19 was present in keratinized or non-keratinized normal tissues [32,51]. This GOE was exhibited in three ways: (i) suprabasal expression of K19 [31,41,46,50,51,52]; (ii) increase in staining intensity [51,52,62]; and (iii) number of samples that expressed K19 in the group [33,36,57]. Further information: Additionally, Fillies [36] found a significant difference of expression between OLK without dysplasia and OLK with dysplasia, with the latter gaining more K19 expression. Four studies [41,46,52,62] showed severe dysplasia or carcinoma in in situ (CIS) samples, which expressed K19 in the full thickness of the epithelium, which was different to mild and moderate lesions (whereby K19 was not found in the superficial cells). However, GOE of K19 was not ubiquitous in all samples [36,52,57,68]. Additionally, increased K19 expression and spread may be seen in inflammatory tissues [50]. Studies that do not support the general trend: The only exception to this were Cema’s [31], Sakamoto’s [53], and Su’s [61] papers. In Cema’s paper [31], there was no expression of K19 in normal and dysplastic samples; hence, no differences were noted. In Sakamoto’s paper [53] and Su’s paper [61], there was LOE of K19 from the basal layer when normal controls were compared to dysplastic samples.	Suprabasal extension of K19 can possibly be utilized to distinguish between OLK with dysplasia and OLK without dysplasia, as well as grades of dysplasia (K19 detection in full thickness of the epithelium may indicate HGD) However, this suprabasal spread may also be seen in inflammatory changes, and as such, it restricts its use only if certain conditions are met; that is, any clinical causes of inflammation needs to be ruled out. Additionally, its diagnostic utility in samples with a high amount of lymphocytic infiltrate needs to be explored further.
K20	2 studies: Sakamoto [53,57,62,63,68] Vaidya [63]	No K20 protein expression in normal and dysplastic samples in Sakomoto’s [53] and Vaidya’s [63] studies.	Limited and poor quality of data available for K20. All studies had small sample sizes and no statistical analysis was completed. Further studies are required to assess if there is any GOE in both protein expression in OLK samples and its implications.

Legend: CIS—carcinoma in situ; GOE—gain of expression; HGD—high-grade dysplasia; LGD—low-grade dysplasia, LOE—loss of expression; OLK—oral leukoplakia.

**Table 4 ijms-25-05597-t004:** General trends of keratin mRNA expression in oral leukoplakia.

**Keratin**	**Studies That Investigated the Keratins of Interest**	**Expression in Oral Leukoplakia (OLK)**	**Overall Impressions for Utility**
K1 and/or K10	1 study in total: Bloor [29] (ISH)	There was decreasing K1 and K10 mRNA expression with worsening grades of dysplasia (mild, moderate, severe).	Decreased K1 and K10 mRNA expression with worsening grades of dysplasia, which is similar to protein expression. May have utility in differentiating LGD from HGD.
K2e	1 study in total: Bloor [30] (ISH)	Expression of K2e was widespread in basal and suprabasal layers in normal epithelium. K2e mRNA is mainly detected in basal and lower suprabasal layers in mild-to-moderate dysplasia but spreading to higher suprabasal layers in moderate–severe dysplasia.	Spread of K2e mRNA expression in higher suprabasal layers in moderate-to-severe dysplasia, which is similar to protein expression. May have utility in differentiating LGD from HGD.
K4 and/or K13	2 studies in total: Bloor [29] (ISH) Ida Yonemochi [39] (ISH)	In Bloor’s [29] study, there was decreasing K4 and K13 mRNA expression with worsening grades of dysplasia (mild, moderate, severe) In Ida Yonemochi’s [39] study, K13 mRNA expressed was reduced in CIS samples as compared to normal controls. This occurred in areas where there was decreased K13 protein expression and no K13 protein expression.	Decreasing K4 and K13 mRNA expression with worsening dysplasia, which is similar to protein expression. May have utility in differentiating LGD from HGD.
K5 and/or K14	2 studies in total: Su [61] (ISH) Yoshida [68] (RT-PCR with electrophoresis)	Su’s paper [61] showed K14 basal and parabasal layer staining in non-keratinized normal tissues, with suprabasal spread of mRNA expression in dysplastic samples. K14 mRNA was found up to the prickle cell layers in normal keratinized tissue. However, no dysplastic samples from keratinized sites acquired for any comparison. In Yoshida’s study [68], electrophoresis expression for K14 increased from normal controls to LGD. However, it progressively decreased from LGD to HGD to OSCC, suggesting decreasing levels of K14 mRNA with worsening dysplasia.	K14 mRNA suprabasal extension in dysplastic samples, which is similar to protein expression. May have utility in detection of dysplasia.
K7	2 studies in total: Shahabinejad [58] (ISH) Su [60] (ISH)	Shahabinejad [58] found K7 mRNA expression in dysplastic tissues. There were no normal controls, and they did not grade dysplasia, as such comparisons between different grades were not completed. However, K7 mRNA increased significantly in their OSCC samples as compared to their OLK samples. Sue [60] found K7 mRNA in the basal and spinous cells of both keratinized and non-keratinized mucosa. Protein was only stained in Merkel cells in keratinized epithelium. In their non-keratinized dysplastic samples, K7 mRNA was stained in all layers in all their samples.	K7 mRNA suprabasal extension in dysplastic samples. May have utility in detection of dysplasia.
K8 and/or K18	1 study in total: Su [60] (ISH)	Su’s study [60] found K8 and K18 mRNA in the basal and lower spinous cells of normal controls. In their dysplastic samples, K8 and K18 mRNA was expressed in all layers, with no suprabasal protein expression.	K8 and K18 mRNA suprabasal extension in dysplastic samples. May have utility in detection of dysplasia.
K16	1 study in total: Ida Yonemochi [39] (ISH)	GOE of K16 mRNA localized in the same areas of K16 protein expression for the 7/23 cases that were positive for K16 on IHC.	K16 mRNA was consistent with K16 protein expression. The utility of this gain of this expression needs to be explored further.
K19	1 study in total: Su [61] (ISH) Yoshida [68] (RT-PCR with electrophoresis)	Su’s study [61] found K19 mRNA expression in basal and parabasal layers in non-keratinized epithelium and localized mainly in the rete ridges of keratinized epithelium. In non-keratinized dysplastic tissues, mRNA expression was detected in suprabasal layers. However, no dysplastic samples from keratinized sites acquired for any comparison. In Yoshida’s study [68], electrophoresis expression for K19 increased from normal to LGD to HGD to OSCC, suggesting increasing levels of K19 mRNA with worsening dysplasia.	K9 mRNA suprabasal extension in dysplastic samples. May have utility in detection of dysplasia.
K20	1 study in total: Shahabinejad [58] (ISH)	Shahabinejad [58] found K20 mRNA expression in dysplastic tissues. There were no normal controls, and they did not grade dysplasia; as such, comparisons between different grades were not completed. However, K20 mRNA increased significantly in their OSCC samples as compared to their OLK samples.	Limited data from which to draw any conclusions. However, it is worth analyzing differences in K20 mRNA expression in various grades of dysplasia given its marked increased in OSCC compared to OLK as a whole.

CIS—carcinoma in situ; HGD—high-grade dysplasia; ISH—in situ hybridization; LGD—low-grade dysplasia; OLK—oral leukoplakia; OSCC—oral squamous cell carcinoma; RT-PCR—reverse transcriptase polymerase chain reaction.

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
