# Peer review of "Unraveling the Keratin Expression in Oral Leukoplakia: A Scoping Review"

_ijms, 2024, doi:10.3390/ijms25115597_

Round 1

Reviewer 1 Report

Comments and Suggestions for Authors

“… This scoping review aims to outline the current knowledge of original studies on human tissues regarding the expression and utility as diagnostic, prognostic and predictive biomarkers in oral leukoplakias …” - The title refers to a systematic review and not a scoping.

 ScienceDirect; is not a research base.

Authors must define inclusion criteria and the filters.

The authors did not perform the research equations. Each database used must have specific equations. Therefore, the methodology cannot be accepted.

Author Response

We thank the reviewer for their time in assisting us in improving the quality of the manuscript.

“… This scoping review aims to outline the current knowledge of original studies on human tissues regarding the expression and utility as diagnostic, prognostic and predictive biomarkers in oral leukoplakias …” - The title refers to a systematic review and not a scoping.

Our manuscript is a scoping review; we changed the title to reflect that. A scoping review presents an overview of a heterogeneous or emerging view topic. The idea of a scoping review is not to present the strength of evidence (hence, unlike a systemic review, there is no assessment of the quality of studies/risk of) but to identify trends and gaps in literature to conduct further research.

 ScienceDirect; is not a research base.

Science Direct is a scientific database published by Elsevier. We chose it because some journals are only available on this database. 

Authors must define inclusion criteria and the filters.

Thanks. This has been revised.

The authors did not perform the research equations. Each database used must have specific equations. Therefore, the methodology cannot be accepted.

Please also look at ‘Supplementary Material 1 - Search Strategy’

Reviewer 2 Report

Comments and Suggestions for Authors

I revised the manuscript “Unraveling the keratin expression in oral leukoplakia: A systematic review”, a systematic review with the aim of showing the current evidence on the diagnostic and prognostic role of biomarkers expressed in oral leukoplakia.

Below I will present some doubts and clarifications divided by section, with the aim of improving the manuscript:

Abstract:

-        Given the centrality of keratin, I would suggest that the authors specify in the aim that the biomarker investigated is keratin.

-        The acronym OLK is not previously explained in the abstract.

-        To better understand the text, I would like to ask the Authors what they mean by "predictive" and what is the implied difference compared to "prognostic".

Introduction:

-        There are several non-previous acronyms explained.

-        Vimentin is first written to be a class III protein, later class II. Double check all classes.

-        “Thorough reviews of the various keratins in different cell types, tissues and locations have been summarized elsewhere”. I think there is no need to specify this, it is normal that the introduction does not include a detailed discussion that is not related to the rationale or purpose of the article. I would suggest the authors eliminate this sentence.

Methods:

-        Table 1. The table is difficult to read. For example, the criterion that only human studies were included is entered into the population. Greater precision in defining the criteria is necessary. For example, were subjects of all genders and ages included? Etc.

For the "loss/gain etc. expression" part of the outcomes, it must clearly specify what.

-        It is not clear what type of biopsy was included, not brush, but which was included? Exclissional incisional of tissue samples, FNAB, etc...?

-        The question of revision is missing.

Since several aspects of the systematic methodology have not been well clarified and followed, I request the Authors to complete the checklist for revisions. So that once the checklist has been compiled and the methodology of the manuscript has been improved, it is easy to revise, as well as provide the Authors themselves with all the points to improve their manuscript.

- Greater precision is also necessary regarding the chronological order in which the revision methodology must be written.

- It must be specified whether the manual search was also conducted or not.

- Data synthesis is missing.

- Risk of bias assessment of included articles is missing.

Results:

-        Check the flow chart, the numbers are not correct (e.g. more than 1,000 records are indicated on scopus alone).

-        The risk of bias must be assessed and commented on.

-        Understanding what the main results are is very difficult from the tables alone as they are 30 pages long overall. It would be advisable to reduce them and be more uniform in their compilation, with less discursive and more concise sentences being tables. Instead, a qualitative presentation in discursive form of the main findings is needed.

Discussion:

-        The authors state: " As such, the level of evidence was difficult to as-certain. However, two variables that directly impact the level of evidence include sample 8 sizes and the completion of statistical analysis (SA).”. Authors should assess the risk of bias of included studies in order to discuss the level of evidence for an included study.

-        Much of the discussion is a concise and written presentation of the results, but more commentary on the main results obtained is necessary. We should reorganize the results and therefore the discussion a little.

Conclusion: The conclusions of a study should not present questions, but rather provide conclusions based on the results obtained. Questions expressed preferably in an indirect form can instead fall into contexts such as future prospects.

References: They are not formatted according to the journal guidelines.

Author Response

I revised the manuscript “Unraveling the keratin expression in oral leukoplakia: A systematic review”, a systematic review with the aim of showing the current evidence on the diagnostic and prognostic role of biomarkers expressed in oral leukoplakia.

Below I will present some doubts and clarifications divided by section, with the aim of improving the manuscript:

Thank you for your comments.

Abstract:

-        Given the centrality of keratin, I would suggest that the authors specify in the aim that the biomarker investigated is keratin.

-        The acronym OLK is not previously explained in the abstract.

  •        To better understand the text, I would like to ask the Authors what they mean by "predictive" and what is the implied difference compared to "prognostic".

Acronyms have been fully listed in the first introduction of the text. The description of prognostic, diagnostic and predictive biomarkers has been outlined in the introduction section. The space in the abstract is not relevant to explain these basic terms. 

Introduction:

  •        There are several non-previous acronyms explained.

Thanks. all checked and corrected.

  •        Vimentin is first written to be a class III protein, later class II. Double check all classes.

Thanks. Checked and corrected.

  •        “Thorough reviews of the various keratins in different cell types, tissues and locations have been summarized elsewhere”. I think there is no need to specify this, it is normal that the introduction does not include a detailed discussion that is not related to the rationale or purpose of the article. I would suggest the authors eliminate this sentence.

Good point. We revised the introduction.  

The utility of keratins in the OLKs may include diagnostic; for example, it may be used as an adjunct for the detection or grading of dysplasia. Keratins may be used as a prognostic tool, for example, in predicting malignant transformation (MT). They may also be utilised as a predictive tool to identify the ideal treatment[22]. Hence, this manuscript aims to review the current knowledge of keratins expressed in OLK and their potential roles (e.g. diagnostic, prognostic or predictive).

Methods:

  •        Table 1. The table is difficult to read. For example, the criterion that only human studies were included is entered into the population. Greater precision in defining the criteria is necessary. For example, were subjects of all genders and ages included? Etc.

Thanks. Revised to improve clarity, but it still appears as a long table.

For the "loss/gain etc. expression" part of the outcomes, it must clearly specify what.

Corrected.

  •        It is not clear what type of biopsy was included, not brush, but which was included? Exclissional incisional of tissue samples, FNAB, etc...?

Missing details have been added.

  •        The question of revision is missing.

Corrected

Since several aspects of the systematic methodology have not been well clarified and followed, I request the Authors to complete the checklist for revisions. So that once the checklist has been compiled and the methodology of the manuscript has been improved, it is easy to revise, as well as provide the Authors themselves with all the points to improve their manuscript.

  • Greater precision is also necessary regarding the chronological order in which the revision methodology must be written.

Thanks. we followed your recommendation. 

  • It must be specified whether the manual search was also conducted or not.

A manual search was used. Additional details have been added.

- Data synthesis is missing.

  • Risk of bias assessment of included articles is missing.

We conducted a scoping review that didn't warrant an assessment of the quality of the included studies. Hence, we didn't perform a risk of bias assessment. 

Results:

  •        Check the flow chart, the numbers are not correct (e.g. more than 1,000 records are indicated on scopus alone).

This has been double checked and revised.

  •        The risk of bias must be assessed and commented on.

See comments above.

-        Understanding what the main results are is very difficult from the tables alone as they are 30 pages long overall. It would be advisable to reduce them and be more uniform in their compilation, with less discursive and more concise sentences being tables. Instead, a qualitative presentation in discursive form of the main findings is needed.

Table 1 is very long because the study included 43 papers. We tried our best to simplify it to improve the reading flow of it. The findings column from Table 2 has been removed as Table 3 consolidates and interprets the data. This has reduced the page count by 7 pages.

Discussion:

-        The authors state: " As such, the level of evidence was difficult to as-certain. However, two variables that directly impact the level of evidence include sample 8 sizes and the completion of statistical analysis (SA).”. Authors should assess the risk of bias of included studies in order to discuss the level of evidence for an included study.

  •        Much of the discussion is a concise and written presentation of the results, but more commentary on the main results obtained is necessary. We should reorganize the results and therefore the discussion a little.

We revised the discussion considerably to move the results to their section.

Conclusion: The conclusions of a study should not present questions, but rather provide conclusions based on the results obtained. Questions expressed preferably in an indirect form can instead fall into contexts such as future prospects.

Thanks. Revised.

References: They are not formatted according to the journal guidelines.

Corrected.

Round 2

Reviewer 1 Report

Comments and Suggestions for Authors

"After analyzing the corrections that the authors introduced in the article, I consider it accepted for publication."

Author Response

Thank you for accepting our manuscript

Reviewer 2 Report

Comments and Suggestions for Authors

-        Acronyms in the abstract must be written in full first, even if the acronym will be written later in the manuscript. Please correct this.

-        I apologize to the Authors if the request for further explanation is not considered valid, as the Authors consider these concepts to be "basic", as indicated in the response provided.

As a reviewer, and as this terminology was used in the aim of the study, I want to understand the difference that the authors implied. These concepts considered "basic" prognostic/predictive were considered synonymous in any studies. I ask for an explanation (not to be added in the abstract) of these "basic terms".

-        I thank the Authors for modifying table 1. It is probably better to have a longer table but with precise inclusion/exclusion criteria. If the authors do not like the long table, the PECOS can also be written in discursive form. Furthermore, as PECOS, the "S" of studies must be written following the Outcomes, and not together with the population. The population is not the type of studies. Finally, is a more specific terminology for "normal tissue" possible? What is meant? Tissue without macroscopic alterations? To distinguish it from "Non dysplastic tissue". And avoid misunderstandings.

-        Not all requests regarding the methodology review have been thoughtful.

The question of study continues to be missing. Information about the publication date/status of included studies, etc. All as indicated in the scoping review checklist.

And also the synthesis of the data and the critical evaluation of the individual sources of evidence (mistakenly called by me first risk of bias). It would be desirable to have the latter and I requested it as the summary of the data was also missing. Therefore, a lot of information on the methodology is missing. Check the PRISMA-ScR checklist.

-        The Authors declare "The reference list was also manually scanned to remove duplicates." Can you explain to me in what sense?

And the manual research that the Authors wrote to me that was conducted where it was stated in the methodology?

-        The flow chart is not correct for the calculations. E.g.: 567+140+1406+21+385 = 2519 records identified. The number of not eligible records is missing (from 568 to 80?).

The study is very interesting, but the methodology is not yet suitable for publishing the article. Unfortunately (or fortunately) this type of study must follow more precise guidelines and every aspect must be clearly specified. Follow the scoping review checklist and the complete manual, point by point.

Author Response

Dear reviewer,

Thank you so much for the invaluable comments on improving our manuscript.

We hope this revision satisfies the required changes. 

Here are below our responses to your comments.

  •        Acronyms in the abstract must be written in full first, even if the acronym will be written later in the manuscript. Please correct this.

Thanks- this has been corrected.

-        I apologize to the Authors if the request for further explanation is not considered valid, as the Authors consider these concepts to be "basic", as indicated in the response provided.

As a reviewer, and as this terminology was used in the aim of the study, I want to understand the difference that the authors implied. These concepts considered "basic" prognostic/predictive were considered synonymous in any studies. I ask for an explanation (not to be added in the abstract) of these "basic terms".

We considered your comments and added details in the introduction to outline the scope of prognostic and predictive biomarkers.

  •        I thank the Authors for modifying table 1. It is probably better to have a longer table but with precise inclusion/exclusion criteria. If the authors do not like the long table, the PECOS can also be written in discursive form. Furthermore, as PECOS, the "S" of studies must be written following the Outcomes, and not together with the population. The population is not the type of studies. Finally, is a more specific terminology for "normal tissue" possible? What is meant? Tissue without macroscopic alterations? To distinguish it from "Non dysplastic tissue". And avoid misunderstandings.

We revised the tables to improve their readability. 

-        Not all requests regarding the methodology review have been thoughtful.

The question of study continues to be missing. Information about the publication date/status of included studies, etc. All as indicated in the scoping review checklist.

And also the synthesis of the data and the critical evaluation of the individual sources of evidence (mistakenly called by me first risk of bias). It would be desirable to have the latter and I requested it as the summary of the data was also missing. Therefore, a lot of information on the methodology is missing. Check the PRISMA-ScR checklist.

-        The Authors declare "The reference list was also manually scanned to remove duplicates." Can you explain to me in what sense?

And the manual research that the Authors wrote to me that was conducted where it was stated in the methodology?

-        The flow chart is not correct for the calculations. E.g.: 567+140+1406+21+385 = 2519 records identified. The number of not eligible records is missing (from 568 to 80?).

The study is very interesting, but the methodology is not yet suitable for publishing the article. Unfortunately (or fortunately) this type of study must follow more precise guidelines and every aspect must be clearly specified. Follow the scoping review checklist and the complete manual, point by point.

We thank the reviewer again. We have addressed the above points as best as possible, including going through the entire PRISMA-SCr checklist and itemizing every point in the article (please see the comments section).

Of note, the following items are not applicable as mentioned in the PRISMA-ScR checklist:
Item 13: summary measures
Item 15: risk of bias across studies
Item 16: additional analyses
Item 22: risk of bias across studies
Item 23: additional analyses

Of note, the following items are optional:
Item 12: critical appraisal of individual sources of evidence
Item 19: critical appraisal within sources of evidence
We would also like to point to the reviewer, that brief mention of weaknesses in individual studies and how they impacted the overall impression was noted in table 3.

However, that is not the main focus of this scoping review.

To quote one of the main papers used to construct the PRIMSA-ScR by Peters et al 2015:

Another distinction between scoping reviews and systematic reviews is that unlike a systematic review, scoping reviews are designed to provide an overview of the existing evidence base regardless of quality. Hence, a formal assessment of methodological quality of the included studies is generally not performed.

Peters, Micah D.J. BHSc, MA(Q), PhD1; Godfrey, Christina M. RN PhD2; Khalil, Hanan BPharm, MPharm, PhD3; McInerney, Patricia PhD4; Parker, Deborah5; Soares, Cassia Baldini RN, MPH, PhD6. Guidance for conducting systematic scoping reviews. International Journal of Evidence-Based Healthcare 13(3):p 141-146, September 2015. | DOI: 10.1097/XEB.0000000000000050

Round 3

Reviewer 1 Report

Comments and Suggestions for Authors

After analyzing the corrections that the authors introduced in the article, I consider it accepted for publication.

Reviewer 2 Report

Comments and Suggestions for Authors

I thank the authors for greatly improving the manuscript and engaging in a healthy discussion with the reviewer.

I believe that the manuscript is currently eligible for publication.

Good luck